# Predictors and Outcomes of SGLT2 Inhibitor Discontinuation in a Real-World Population after Hospitalization for Heart Failure

**DOI:** 10.3390/biomedicines11030876

**Published:** 2023-03-13

**Authors:** Masaki Nakagaito, Teruhiko Imamura, Ryuichi Ushijima, Makiko Nakamura, Koichiro Kinugawa

**Affiliations:** Second Department of Internal Medicine, University of Toyama, Toyama 930-0152, Japan

**Keywords:** cardiovascular disease, heart failure, chronic complications, type 2 diabetes

## Abstract

Background: Sodium–glucose cotransporter 2 inhibitors (SGLT2i) reduce mortality and morbidity in patients with heart failure (HF), but are discontinued in some patients. Such patients may not enjoy favorable benefits of SGLT2i therapy. We evaluated the risk factors for SGLT2i discontinuation in a real-world population with HF. Methods: We retrospectively included consecutive patients who were hospitalized for HF and administered SGLT2i during the index hospitalization between February 2016 and September 2021. We assessed the baseline clinical factors associated with post-discharge discontinuation of SGLT2i. Results: This study included a total of 159 patients (median age = 73 years, 57 women). Among baseline characteristics, a lower serum albumin level (odds ratio = 0.23, 95% confidence interval = 0.07–0.76, *p* = 0.016) and a higher dose of furosemide (odds ratio = 1.02, 95% confidence interval = 1.00–1.05, *p* = 0.046) were independently associated with the future discontinuation of SGLT2i following index discharge. Patients who terminated SGLT2i (*n* = 19) had a higher incidence of HF recurrence or cardiovascular death during the 1-year therapeutic period (32% versus 11%, *p* = 0.020). Conclusions: Among patients who initiated SGLT2i during hospitalization for HF, lower serum albumin levels and higher doses of loop diuretic at index discharge were associated with the discontinuation of SGLT2i following index discharge. We should pay special attention to patients with such characteristics during the initiation of SGLT2i and during SGLT2i therapy.

## 1. Introduction

To date, sodium–glucose cotransporter 2 inhibitors (SGLT2i) have demonstrated a major therapeutic advance in patients with heart failure (HF). Large-scale randomized controlled trials (RCTs) have shown that they reduce the risk of hospitalization for HF or cardiovascular death in patients with HF with reduced left ventricular ejection fraction (HFrEF) [1,2]. Thus, dapagliflozin and empagliflozin, an example of SGLT2i, have received a class I indication for HFrEF patients in international guidelines [3,4]. The EMPEROR-Preserved trial with empagliflozin and DELIVER trial with dapagliflozin have shown a reduction in HF hospitalization in patients with HF with preserved left ventricular ejection fraction (HFpEF) [5,6]. Furthermore, a reduction in HF hospitalization was reported in patients with HF, independently of the presence of diabetes mellitus (DM). Thus, SGLT2i are prescribed in patients with chronic HF accompanying a wide range of values of left ventricular ejection fraction (LVEF). Recently, several RCTs suggested that SGLT2i might exert a beneficial effect on the cardiovascular outcome even in hospitalized or recently discharged patients with HF, in addition to those with chronic HF [7,8,9].

However, our team previously found that some studies that initiated SGLT2i terminated SGLT2i for a variety of reasons [10,11]. Patients who terminated SGLT2i had worse clinical outcomes compared with those who continued SGLT2i. Thus, we should pay attention to the negative aspect of SGLT2i therapy. Given their pleiotropic effect, several adverse drug reactions including genital and urinary tract infection [12], volume depletion [13], diabetic ketoacidosis (DKA) [14], bone fractures [15], and amputation [16] might be increased.

Previous large-scale RCTs demonstrated that there was no notable excess of serious treatment-emergent adverse events during SGLT2i therapy. Nevertheless, minor adverse effects of SGLT2i treatment are potentially numerous, and real-world data that reflect actual practice should be useful in identifying them. Of note, although clinical implication of early administration of SGLT2i was investigated in several RCTs including the SOLOIST-WHF trial and EMPULSE trial, whether early-administered SGLT2i can be continued over a longer observational period remains uncertain.

An abundant number of comorbidities including HF and other baseline characteristics may affect the extent of SGLT2i treatment and disturb continuation of SGLT2i therapy. Such risk factors associated with the termination of SGLT2i are of great importance for clinicians in determining whether to initiate SGLT2i and managing patients receiving SGLT2i over a long-term therapeutic period. In the present study, we investigated the factors associated with post-discharge discontinuation of SGLT2i, which was initiated during index hospitalization soon after the stabilization of HF.

## 2. Materials and Methods

This was a single-center, retrospective observational study to investigate the baseline factors associated with the discontinuation of SGLT2i, which was initiated during index hospitalization. This study was carried out in accordance with the principles outlined in the Declaration of Helsinki, and the institutional ethics board of Toyama University Hospital approved the study protocol (#R2015154, approval date 11 April 2016). Written informed consent was obtained from all of the participants beforehand.

### 2.1. Study Population

Patients who had been admitted for HF, which was diagnosed according to the Framingham criteria, at our institute between April 2016 and September 2021 were included in this study. Most of the patients had New York Heart Association (NYHA) class III/IV symptoms upon admission. All patients were treated with guideline-directed medical therapy for HF, including renin–angiotensin system inhibitors or angiotensin receptor–neprilysin inhibitors, beta-blockers, mineralocorticoid receptor antagonists, and diuretics, if applicable. Patients who newly received SGLT2i during their index hospitalization immediately following the stabilization of hemodynamics were followed-up for 12 months following index discharge (defined as day 0).

### 2.2. Exclusion Criteria

We excluded the following patients: age < 20 years, end-stage renal failure with estimated glomerular filtration rate (eGFR) < 15 mL/min/1.73 m^2^, use of durable left ventricular assist devices, pregnancy or breastfeeding, and current use of SGLT2i during index hospitalization. Patients who stopped taking SGLT2i during index hospitalization were excluded. Adjustment of medical therapy was permitted as a real-world clinical practice. Patients who were lost to follow-up during the 12-month observational period were also excluded. Patients who died due to non-cardiovascular disease were excluded.

### 2.3. Study Design and Data Collection

Baseline characteristics including demographics and laboratory data at index discharge were retrieved. The eGFR was calculated using the guidelines from the Chronic Kidney Disease Epidemiology Collaboration. Plasma B-type natriuretic peptide (BNP) level and eGFR were retrospectively retrieved at the 12-month follow-up. Standard echocardiographic findings during index hospitalization were retrieved. For the present analysis, participants were divided into HFrEF (LVEF < 40%), HfmrEF (LVEF 40–49%), and HfpEF (LVEF ≥ 50%) groups. We defined DM as patients satisfying glycated hemoglobin (HbA1c) ≥ 6.5% or receiving antidiabetic treatment. Bacterial infections included symptomatic bacteriuria or cases where bacterial cultures from sources such as blood, urine, or stool were positive. When patients died from cardiovascular causes, they were censored at the time of events.

### 2.4. Study Endpoints

The primary outcome was a discontinuation of SGLT2i during the 12 months following their index discharge. Baseline characteristics that would significantly affect the primary outcome were investigated. The secondary outcome was the changes in plasma BNP level and eGFR during the observational period.

### 2.5. Statistical Analyses

Statistical analysis was conducted using JMP^®^ 15.0 (SAS Institute Inc., Cary, NC, USA). The level of significance was set at two-tailed *p* < 0.050. Continuous variables are presented as the median and interquartile range unless specifically stated. Categorical variables are presented as absolute numbers and percentages. The Wilcoxon test was used to compare continuous parameters, and Pearson’s χ^2^ test was used for comparison of categorical variables.

Univariable and multivariable analyses with logistic regression models were conducted to calculate the adjusted odds ratio to assess the influence of various baseline parameters on the discontinuation of SGLT2i. Variables deemed as significant (*p* < 0.050) in the univariate analyses were enrolled in the multivariate analyses. Time-to-event outcomes were evaluated using Kaplan–Meier estimates and the two treatment groups were compared with the use of log-rank test statistics.

## 3. Results

### 3.1. Follow-Up and Patient Characteristics

Totally, 187 patients met the inclusion criteria. Of them, 2 died from non-cardiovascular causes, and 26 patients were lost to follow-up during the observational period. Finally, 159 patients were included in this study (Figure 1).

Table 1 lists the baseline characteristics obtained at index discharge. The median age was 73 (64–81) years and 36% of patients were women. NYHA class III/IV symptoms were noted in 36 patients (23%). HFrEF was noted in 62 patients (39%), HFmrEF in 42 patients (26%), and HFpEF in 55 patients (35%). DM was noted in 123 patients (77%) (all of them were type 2). As for the types of SGLT2i, 37 patients received canagliflozin, 82 received dapagliflozin, and 40 received empagliflozin. All patients receiving canagliflozin had a history of DM.

### 3.2. Causes of Primary Outcome

As a primary outcome, a total 19 of patients (12.0%) encountered the discontinuation of SGLT2i. The most common reasons for SGLT2i discontinuation were bacterial infection (*n* = 6, 32%), followed by patient-reported side effects (*n* = 5, 26%). The side effects reported by patients included nocturia, anorexia, pruritus, and tiredness. Adjustment for treatment of cardiovascular disease was the third most common reason (*n* = 4, 21%) (Figure 2).

### 3.3. Prediction of Primary Outcome

Baseline characteristics stratified by achievement of primary endpoint are summarized in Table 1. The level of serum albumin was lower in the discontinuation group than the continuation group. Medications for HF and DM were not statistically different between the two groups. There were no significant differences in the plasma BNP and eGFR levels at baseline between both groups.

Univariable and multivariable analyses demonstrated that serum albumin level (odds ratio = 0.23, 95% confidence interval = 0.07–0.76, *p* = 0.016) was independently and negatively associated with the discontinuation of SGLT2i, and furosemide dosage (odds ratio = 1.02, 95% = confidence interval 1.00–1.05, *p* = 0.046) was positively associated with the discontinuation of SGLT2i (Table 2). Prevalence of DM was not associated with the primary outcome.

### 3.4. Impact of SGLT2i Discontinuation on Clinical Outcomes

Patients in the discontinued SGLT2i group had a higher incidence of recurrent HF hospitalization or cardiovascular death (32% vs. 11%, *p* = 0.020; Figure 3).

The changes in plasma BNP and eGFR were assessed in 117 and 133 patients, respectively, by excluding those who died during the observational period and those whose laboratory data were not retrieved. Changes in plasma BNP level from baseline are shown in Figure 4. Plasma BNP levels decreased in patients with both continuation and discontinuation of SGLT2i from baseline to 12 months. The changes in plasma BNP were not different between the continued SGLT2i group and the discontinued SGLT2i group (−20.5 vs. −96.0 pg/mL, *p* = 0.096). On the other hand, there were no significant changes in eGFR throughout the 12-month observation period in both groups. The changes in eGFR were not different between the continued SGLT2i group and the discontinued SGLT2i group (+0.65 vs. −0.60 mL/min/1.73 m^2^, *p* = 0.767).

## 4. Discussion

We investigated the factors associated with post-discharge discontinuation of SGLT2i in patients with HF. SGLT2i were initiated soon after the stabilization of hemodynamics during index hospitalization for HF. Among baseline characteristics at index discharge, a lower serum albumin level and a higher dose of furosemide were associated with discontinuation of SGLT2i following index discharge. Patients with discontinued SGLT2i had a higher incidence of re-hospitalization for HF or cardiovascular death during the 12-month observational period following index discharge. The changes in plasma BNP and eGFR were not significantly different between the discontinued SGLT2i group and the continued SGLT2i group.

### 4.1. Discontinuation of SGLT2i

In this study, SGLT2i were discontinued for reasons other than death in 19 patients (12.0%). Our discontinuation rate of 12.0% is similar to previously reported rates of 10.5–23.2% in large-scale RCTs [1,2,5,6]. However, the median follow-up durations of these RCTs (16.0–27.9 months) were longer than ours (12.0 months), and the risk of SGLT2i discontinuation in this study might be higher than in previous RCTs. In the SOLOIST-WHF trial and EMPULSE trial, both of which investigated the clinical implication of early initiation of SGLT2i, the rates of adverse events leading to the discontinuation of SGLT2i were lower than those in the present study (8.5% and 4.8%, respectively) [7,8]. These findings suggest difficulty in continuing SGLT2i in real-world populations.

A possible explanation for the high frequency of SGLT2i discontinuation was that our patients were included in this study soon after an episode of decompensated HF. Our population might be distinct from patients with chronic HF in most of the previous trials, since our patients were at a higher risk of cardiovascular events compared to patients in many of the other trials. Additionally, our population may have a higher risk of SGLT2i withdrawal in the vulnerable period, in which patients may have unstable volume status, renal function, and blood pressure and other HF therapies may require adjustment. Similarly, a subset of the DELIVER trial, which evaluated the response to dapagliflozin in hospitalized or recently discharged patients, reported that serious adverse events were more common in recently hospitalized patients compared with those without episodes of recent hospitalization [17].

### 4.2. Diabetes and SGLT2i Discontinuation

Importantly, prevalence of DM was no longer a predictor of SGLT2i discontinuation in this study. Further, no patients stopped SGLT2i because of major hypoglycemia or DKA. Our observation regarding the safety of SGLT2i is consistent with previous trials and reports that the rate of SGLT2i discontinuation is not significantly different between participants with and without diabetes.

### 4.3. Hypoalbuminemia and SGLT2i Discontinuation

We demonstrated that a lower serum albumin level and a higher dose of furosemide were associated with SGLT2i discontinuation. It is worth noting that both low serum albumin levels and high doses of furosemide at baseline are strongly associated with an increased risk of HF [18,19,20]. In addition, SGLT2i therapy for patients with malnutrition might not be encouraged due to its potential to progress catabolism. Although subsets of the DAPA-HF trial and DELIVER trial demonstrated that dapagliflozin improved cardiovascular outcomes regardless of frailty status [21,22], few studies assessed the efficacy and safety of SGLT2i in malnourished patients. Our finding suggests that caution should be exercised in using SGLT2i in patients with both HF and malnutrition.

### 4.4. High Dose of Loop Diuretics and SGLT2i Discontinuation

High incidence of SGLT2 discontinuation in patients with high doses of furosemide may be associated with adverse events of volume depletion. Although individual RCTs reported that the frequency of adverse events related to volume depletion did not differ between patients with SGLT2i and those without SGLT2i, a meta-analysis indicated that there is an increase in hypovolemia associated with SGLT2i treatment [13]. Moreover, a previous study reported that SGLT2i therapy resulted in a significant augmentation of natriuresis when combined with loop diuretics [23]. Inhibition of compensatory absorption of sodium in the Henle’s loop or distal tubule by loop diuretics in patients receiving SGLT2 may explain the findings of the study.

Therefore, co-administration of SGLT2i and high-dose loop diuretics might act complementarily via double blocking of proximal tubules and Henle’s loop, resulting in rapid volume depletion. This diuretic profile offers significant advantages in the management of volume status in patients with HF, whereas a robust diuretic response may result in excessive fluid loss. In this study, unexpected changes in plasma volume caused by co-administration of SGLT2i and high-dose loop diuretics may have led to SGLT2i discontinuation.

### 4.5. Outcomes of SGLT2i Discontinuation

In our study, patients who discontinued SGLT2i had a higher risk of re-hospitalization for HF or cardiovascular death. This result should be interpreted with caution since this is a retrospective observational study. We found that lower serum albumin levels and higher doses of furosemide were independently associated with SGLT2i discontinuation, and were also associated with an increased risk of HF. Patients who discontinued SGLT2i may be at higher risk of cardiovascular events than those who continued SGLT2i, making it difficult to assess the association between SGLT2i discontinuation and cardiovascular outcome in this study.

On the other hand, we previously reported that patients who discontinued SGLT2i at discharge from hospitalization for HF were more likely to be re-hospitalized for HF than those who continued SGLT2i [11]. Previous findings may support a direct and independent prophylactic effect of SGLT2i on cardiovascular events for post-discharge patients in this study. RCTs demonstrated that SGLT2i improved cardiovascular outcomes, whereas few studies reported the effect of discontinuing SGLT2i on cardiovascular events. Additional research to confirm these findings and to understand the importance of continuing SGLT2i after discharge from hospitalization for HF is needed.

### 4.6. Limitations

This study has several limitations. First, this study was conducted retrospectively in a single center, and both overall sample size and number of patients reaching the primary outcome were small. Given the low event number, the number of potential confounders included in the multivariate analyses was limited. Similarly, limited patients were included in the analysis for secondary outcome. Furthermore, it is not possible to establish a cause–effect relationship given the retrospective observational study design. Second, other HF medications were also adjusted during the observational period in this study. Thus, it is challenging to assess the effect of SGLT2i alone. Lastly, our population received multiple types of SGLT2i in the present study. No RCTs have evaluated the efficacy of canagliflozin in patients with HF. It remains unclear whether such beneficial effects are consistent across individual SGLT2i. However, a recent retrospective cohort study reported that there was no significant difference in the risk of cardiovascular events including HF among patients taking dapagliflozin, canagliflozin, and empagliflozin [24].

## 5. Conclusions

In this study, we found that 12.0% of patients who initiated SGLT2i soon after the stabilization of their HF during index hospitalization terminated SGLT2i therapy within 12 months following index discharge. The rate of discontinuation of SGLT2i is slightly higher than those of previous RCTs. The most common reason for the discontinuation of SGLT2i was bacterial infection. Lower serum albumin levels and higher doses of furosemide at the time of index discharge were independently associated with SGLT2 discontinuation after discharge from hospitalization for HF. Furthermore, SGLT2i discontinuation was associated with an increased risk of re-hospitalization for HF or cardiovascular death. Special attention should be paid when initiating and managing SGLT2i for those with such risk factors.

## Figures and Tables

**Figure 1 biomedicines-11-00876-f001:**
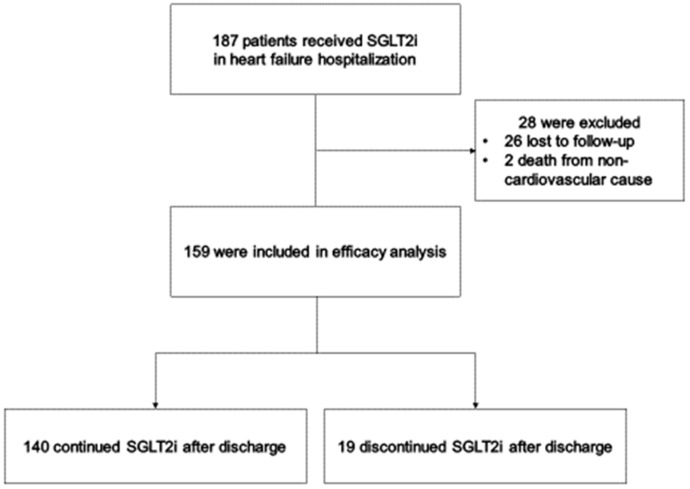
Enrollment and Follow-up.

**Figure 2 biomedicines-11-00876-f002:**
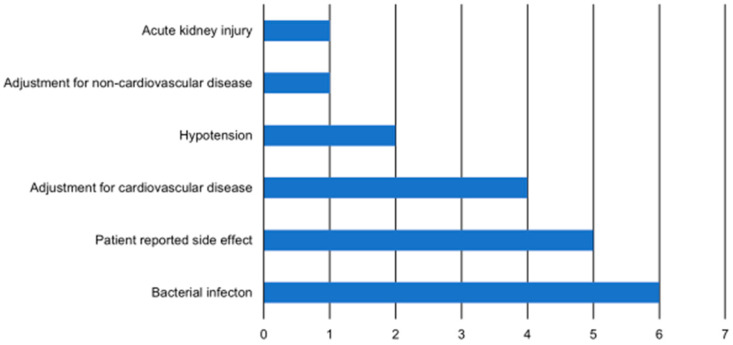
Reasons for discontinuation of SGLT2 inhibitors (*n* = 19).

**Figure 3 biomedicines-11-00876-f003:**
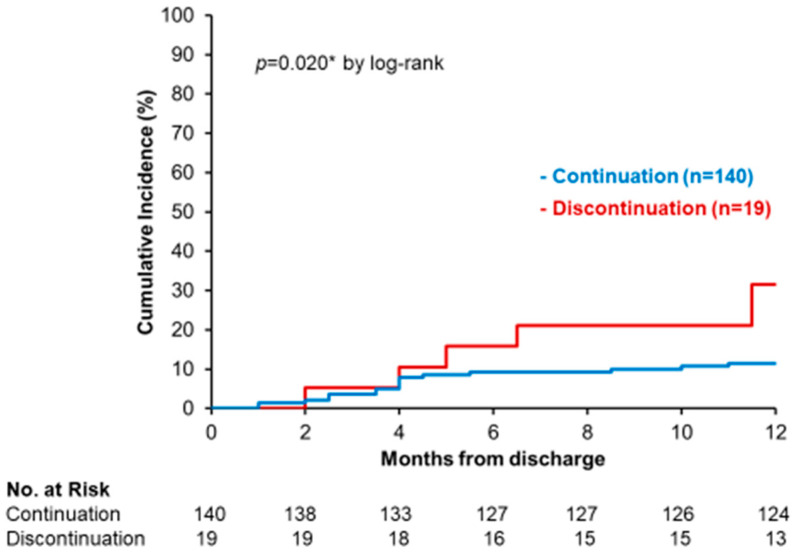
Recurrent hospitalization for heart failure or cardiovascular death in patients with continued SGLT2 inhibitors versus those with discontinued SGLT2 inhibitors. * *p* < 0.050.

**Figure 4 biomedicines-11-00876-f004:**
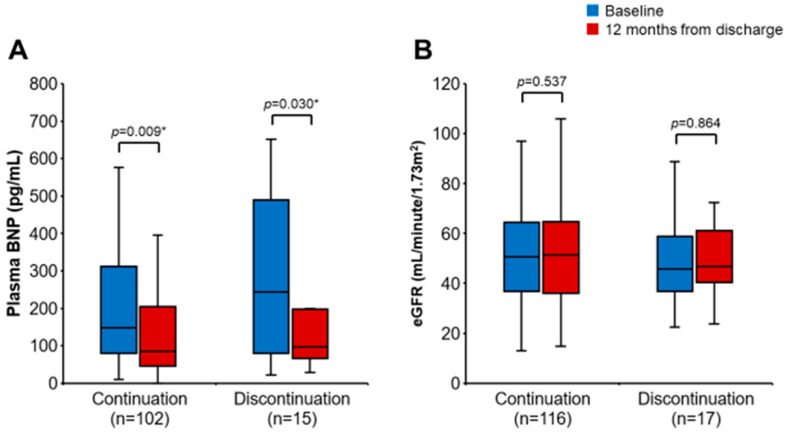
Changes in plasma BNP (**A**) and estimated GFR (**B**) during 12-month observational period following index discharge. Variables were compared by Wilcoxon singed-rank test. * *p* < 0.050.

**Table 1 biomedicines-11-00876-t001:** Baseline characteristics at index discharge.

	Total (*n* = 159)	Continuation (*n* = 140)	Discontinuation (*n* = 19)	*p* Value
Age, years	73 (64–81)	72 (63–81)	75 (69–83)	0.281
Male, *n* (%)	102 (64)	93 (66)	9 (47)	0.104
Body weight, kg	57.3 (50.0–66.8)	57.3 (50.2–67.5)	58.7 (47.0–64.1)	0.547
Body mass index, kg/m^2^	22.6 (19.8–24.9)	22.6 (19.9–24.9)	22.2 (19.8–23.7)	0.720
Systolic blood pressure, mmHg	105 (96–117)	104 (95–117)	108 (98–118)	0.227
Heart rate, beats per minutes	70 (63–78)	70 (63–78)	71 (65–78)	0.655
Diabetes mellitus, *n* (%)	123 (77)	105 (75)	18 (95)	0.054
Ischemic etiology, *n* (%)	66 (42)	58 (41)	8 (42)	0.955
Atrial fibrillation, *n* (%)	45 (28)	37 (26)	8 (42)	0.155
Implantable cardioverter-defibrillator, *n* (%)	23 (13)	22 (16)	1 (5)	0.224
Cardiac resynchronization therapy, *n* (%)	16 (10)	2 (11)	14 (10)	0.943
New York Heart Association class III–IV, *n* (%)	36 (23)	29 (21)	7 (37)	0.115
Left ventricular ejection fraction, %	43 (33–55)	43 (33–55)	43 (31–60)	0.744
Value of <40% (HFrEF), *n* (%)	62 (39)	54 (39)	8 (42)	0.767
Value of 40–49% (HFmrEF), *n* (%)	42 (26)	37 (26)	5 (26)	0.992
Value of ≥50% (HFpEF), *n* (%)	55 (35)	49 (35)	6 (32)	0.769
HbA1c, %	6.7 (6.4–7.6)	6.7 (6.3–7.6)	6.9 (6.5–7.6)	0.404
Fasting blood sugar, mg/dL	110 (95–132)	110 (95–130)	121 (93–152)	0.230
Hemoglobin, g/dL	12.5 (11.2–13.9)	12.6 (11.2–14.1)	11.9 (10.6–13.5)	0.089
Hematocrit, %	37.8 (34.1–41.4)	38.0 (34.2–41.6)	35.5 (32.2–40.6)	0.135
Serum albumin, g/dL	3.6 (3.4–3.9)	3.7 (3.4–4.0)	3.4 (3.2–3.6)	0.006 *
Serum sodium, mEq/L	138 (136–140)	138 (137–140)	136 (135–140)	0.258
Serum potassium, mEq/L	4.3 (4.1–4.6)	4.3 (4.1–4.6)	4.5 (3.9–4.8)	0.641
eGFR, mL/minute/1.73 m^2^	50.5 (36.9–64.4)	51.2 (36.9–66.6)	41.2 (36.9–58.8)	0.342
Uric acid, mg/dL	5.7 (4.8–6.9)	5.7 (4.8–7.0)	6.0 (5.6–6.7)	0.418
Plasma BNP, pg/mL	142 (69–284)	142 (69–269)	182 (68–477)	0.237
Heart failure therapies				
Beta-blockers, *n* (%)	142 (89)	124 (89)	18 (95)	0.415
ACEI/ARB/ARNI, *n* (%)	149 (94)	130 (93)	19 (100)	0.229
Loop diuretics, *n* (%)	102 (64)	88 (63)	14 (74)	0.356
Dose of furosemide, mg/day	10 (0–20)	10 (0–20)	20 (0–20)	0.223
MRA, *n* (%)	54 (34)	44 (31)	10 (53)	0.067
Thiazides, *n* (%)	3 (2)	3 (2)	0 (0)	0.520
Anti-diabetic agents				
Sulfonylureas, *n* (%)	6 (4)	6 (4)	0 (0)	0.358
DPP-4i, *n* (%)	62 (39)	53 (38)	9 (47)	0.425
Biguanides, *n* (%)	22 (14)	21 (15)	1 (5)	0.249
Insulin, *n* (%)	13 (8)	10 (7)	3 (16)	0.197
Sodium–glucose cotransporter 2 inhibitors				
Canagliflozin, *n* (%)	37 (23)	32 (23)	5 (26)	0.738
Dapagliflozin, *n* (%)	82 (52)	75 (54)	7 (37)	0.171
Empagliflozin, *n* (%)	40 (25)	33 (24)	7 (37)	0.211

eGFR, estimated glomerular filtration rate; HFrEF, heart failure with reduced ejection fraction (ejection fraction < 40%); HFmrEF, heart failure with mildly reduced ejection fraction (ejection fraction 40–49%); HFpEF, heart failure with preserved ejection fraction (ejection fraction ≥ 50%); HbA1c, glycated hemoglobin; BNP, B-type natriuretic peptide; NT-proBNP, N-terminal pro-B-type natriuretic peptide; ACEI, angiotensin-converting enzyme inhibitors; ARB, angiotensin receptor blockers; ARNI, angiotensin receptor–neprilysin inhibitors; MRA, mineralocorticoid receptor antagonists; DPP-4i, dipeptidyl peptidase-4 inhibitors. * *p* < 0.050.

**Table 2 biomedicines-11-00876-t002:** Variables associated with discontinuation of SGLT2 inhibitors.

	All Patients (*n* = 159)					
	Univariable Analysis			Multivariable Analysis		
Variables	Odds Ratio	95% CI	*p* Value	Odds Ratio	95% CI	*p* Value
Age, years	1.03	0.98–1.08	0.224			
Male, yes	0.46	0.17–1.20	0.110			
Body mass index, kg/m^2^	0.97	0.86–1.10	0.657			
Systolic blood pressure, mmHg	1.02	0.99–1.05	0.136			
Heart rate, bpm	1.01	0.97–1.06	0.578			
Ischemic etiology, yes	1.03	0.39–2.71	0.955			
Atrial fibrillation, yes	2.03	0.73–5.40	0.161			
NYHA class III–IV, *n* (%)	2.23	0.81–6.18	0.122			
HFrEF, yes	1.16	0.44–3.06	0.767			
Diabetes mellitus, yes	6.00	0.77–46.59	0.087			
Fasting blood sugar, mg/dL	1.01	1.00–1.02	0.219			
Hematocrit, %	0.93	0.85–1.03	0.157			
Serum albumin, g/dL	0.19	0.06–0.62	0.006 *	0.23	0.07–0.76	0.016 *
Serum sodium, mEq/L	0.96	0.84–1.10	0.551			
Serum potassium, mEq/L	1.24	0.47–3.29	0.670			
eGFR, mL/min/1.73 m^2^	0.99	0.97–1.02	0.605			
Uric acid, mg/dL	1.07	0.81–1.42	0.638			
ln BNP	1.41	0.85–2.33	0.187			
Beta-blockers, yes	2.32	0.29–18.59	0.427			
ACEI/ARB/ARNI, yes	NA	NA	0.993			
Loop diuretics, yes	1.66	0.56–4.86	0.360			
Dose of furosemide, mg/day	1.03	1.01–1.05	0.013 *	1.02	1.00–1.05	0.046 *
MRA, yes	2.42	0.92–6.39	0.073			
Thiazides, yes	NA	NA	0.991			
Sulfonylureas, yes	NA	NA	0.992			
DPP-4i, yes	1.48	0.56–3.87	0.427			
Biguanides, yes	0.32	0.04–2.49	0.273			
Insulin, yes	2.44	0.61–9.79	0.209			

NYHA, New York Heart Association; HFrEF, heart failure with reduced ejection fraction (ejection fraction < 40%); LVEF, left ventricular ejection fraction; HbA1c, glycated hemoglobin; eGFR, estimated glomerular filtration rate; BNP, B-type natriuretic peptide; ACEI, angiotensin-converting enzyme inhibitors; ARB, angiotensin receptor blockers; ARNI, angiotensin receptor–neprilysin inhibitors; MRA, mineralocorticoid receptor antagonists; NA, not applicable. * *p* < 0.050.

## Data Availability

The data presented in this study are available on request from the corresponding author. The data are not publicly available due to privacy.

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
