# Peer review of "Predictors and Outcomes of SGLT2 Inhibitor Discontinuation in a Real-World Population after Hospitalization for Heart Failure"

_biomedicines, 2023, doi:10.3390/biomedicines11030876_

Round 1

Reviewer 1 Report

This review of the reasons for discontinuing a SGLT2i in patients hospitalized for HF is of potential importance. However, while you state that you have included patients hospitalized for HF you have omitted several larger studies from your analysis that should be included . These include the Empulse study evaluating the effect of empagliflozin in patients admitted with acuter or worsening HF and randomized to empa prior to hospital discharge  , another study of Empa in patients with acute HF , the subset of the Deliver study  hospitalized for HF and randomized prior to hospital discharge as well as the Soloist study of sotagliflozin  in patients randomized after an episode of hospitalization for worsening HF .  . I would suggest you include these patients in your analysis . 

Reviewer 2 Report

The authors should state why a 12-month follow-up time was chosen. 

I find it not surprising to see that the two variables, which were statistically different between the continuation group and the discontinuation group were also the ones with significant odds ratios in the multivariable model. This likely suggests a statistical artifact and less likely a clinical phenomenon. 

Reviewer 3 Report

Dear Authors,

Please improve and extent your introduction

Please add more conclusion to your article

Kind regards

Reviewer 4 Report

Regarding the manuscript the following comments should be mentioned:

The main concerns regarding the present study are that the consequences of interrupting SGLT2i in patients with heart failure are well known; the number of patients that stopped administering the treatment is quite low in comparison with the cohort and the results might have been impaired; the author previously published an article containing almost the same number of subjects and did not mention it in the references; the Discussion chapter could be expanded. 

Other minor comments: keywords are not representative of the investigated study parameters; "bacterial infection" is a broad term and could be explained; canagliflozin is included in the analysis, but there is no official indication for heart failure; English spelling should be revised. 

Round 2

Reviewer 1 Report

Thank you for responding to my comments

Reviewer 2 Report

N/A

Reviewer 4 Report

The number of patients that discontinued the treatment quite small to draw a conclusions regarding their characteristics in comparison with peers that continued the iSGLT2. The conclusion chapter has been revised, but contain data from other studies or it is not clear (Abstract).